# Reproductive Apparatus, Gonadic Maturation, and Allometry of *Cyclocephala barrerai* Martínez (Coleoptera: Melolonthidae: Dynastinae)

**DOI:** 10.3390/insects13070638

**Published:** 2022-07-16

**Authors:** Abraham Sanchez-Cruz, Daniel Tapia-Maruri, Alfredo Jiménez-Pérez

**Affiliations:** 1Laboratorio de Ecología Química de Insectos, Centro de Desarrollo de Productos Bióticos, Instituto Politécnico Nacional, CEPROBI # 8, San Isidro, Yautepec 62731, Mexico; asanchezc1700@alumno.ipn.mx; 2Laboratorio de Microscopia Avanzada, Centro de Desarrollo de Productos Bióticos, Instituto Politécnico Nacional, CEPROBI # 8, San Isidro, Yautepec 62731, Mexico; dmaruri@ipn.mx

**Keywords:** beetles, reproductive system, sexual dimorphism, oogenesis

## Abstract

**Simple Summary:**

*Cyclocephala barrerai* larvae feed on the roots of ornamental and commercial grasses. Most of their biology is still unknown, and there are no tools to manage their populations. The objective of this work was to study the reproductive system, gonadic maturation, and allometry of *C. barrerai*. This is the first report into the Melolonthidae family that characterizes the reproductive apparatus, gonadic maturation of both sexes, and allometric relationships of virgin F1 specimens. Adults do not feed, so gonadic maturation depends on larvae reserves. Females present sclerotized accessory glands and a genital chamber with muscular fibers and bacteria. The morphology of the parameres and the antennae are under sexual selection. This is the first report of two main differences among sexes: females are heavier while males have longer antennae. Males and females exhibit allometric relationships that can be used to predict the bodyweight of field-collected specimens.

**Abstract:**

The Order Coleoptera provides good examples of morphological specializations in the reproductive apparatus, gonadic maturation, and allometry differing between the sexes. The female and male reproductive apparatus has been modified to ensure reproduction between individuals of the same species. The genus *Cyclocephala* has more than 500 species distributed in America, and *Cyclocephala barrerai* Martínez is an economically important species in the central part of Mexico. The objective of this work was to study the reproductive system, gonadic maturation, and allometry of *C. barrerai*. We used light, scanning electron, and laser scanning confocal microscopy to describe the reproductive apparatus and gonadic maturation of females and males. The relationship between adult weight and different parts of the body was established by linear regression. Regardless, the reproductive apparatuses of *C. barrerai* are like those of other Melolonthidae: the genital chamber, the type II accessory glands, and the ventral plaques of the female and the ejaculator bulb and genital capsule of the males are specific to *C. barrerai*. The gonads are fully developed when 18 d old. The weight of adult *C. barrerai* has a positive linear relationship with distinct parts of its body, while the antennae of males are larger than those of the females.

## 1. Introduction

Gonadic maturation indicates the beginning of the insect’s reproductive period [1,2,3,4]. In families such as Carabidae, Coccinellidae, Curculionidae, Dryophthoridae, Scolytidae, Silphidae, and Chrysomelidae, gonadic maturation is mainly affected by genetics, hormones (produced by the female or obtained from a male during mating), age, temperature, and food intake as larvae and adults [5,6,7].

The male Melolonthidae reproductive apparatus typically comprises: (A) two accessory glands; (B) two vasa deferentia; (C) two testes, each with six follicles connected by the vasa efferentia; (D) an ejaculatory duct; (E) a genital capsule; and (F) an aedeagus [8,9]. The genital capsule with two plaques covers the endophallus. The most distal plaques of the genital chamber are called parameres, which are used by the male to secure the female and guide the aedeagus through the female’s urogenital pore [10]. In Melolonthidae, these parameres are species-specific and have taxonomic and evolutionary significance [11,12].

The female Melolonthidae reproductive apparatus typically has (A) two compound ovaries with six ovarioles each, (B) a common oviduct, (C) a *bursa copulatrix*, (D) a spermatheca with a spermathecal gland, (E) a genital chamber, (F) one or two accessory glands, and (G) ventral plaques [8,13,14].

Information on female Melolonthidae gonadic maturation is scarce. Yi-Zhen et al. [15] distinguished three types of female gonadic maturation: (1) those females that emerge with fully developed ovaries, such as *Holotrichia* sp., *Holotrichia trichopora* Fairmaire, and *Anomala exoleta* Fald; (2) those females that require some food after emergence to fully develop their gonads, such as *Holotrichia oblita* Faldermann and *Anomala corpulenta* Motschulsky; and (3) females whose gonadic maturation entirely depends on food intake as adults, such as *Holotrichia parallela* Motschulsky. The only information available about Melolonthidae ovary type is from the New Zealand grass grub *Costelytra zealandica* White with a telotrophic ovary [16].

Gayon [17] defined allometry as “the changes in relative dimensions of parts of an organism that are correlated with changes in overall size”. The allometry of specific morphological characters is important in Coleoptera reproduction as those characters are relevant during courtship, mate selection, mating, coercive behavior, and female manipulation during sperm transfer [18,19,20,21]. Most of the information on allometry in Melolonthidae is in the Dynastinae subfamily, where males show a positive allometric relationship between size and horns; larger males have larger horns, providing an advantage to larger males during agonistic encounters and courtship [22,23,24]. The claw of the first leg of males of *Cyclocephala* is longer in males than in females [25], but its role during courtship and mating is unknown.

Despite the relevance of the genus *Cyclocephala* as pollinators or pests of different economically important crops [26,27,28], the morphology of their reproductive apparatus is not fully studied. The genital chamber and accessory glands of *Cyclocephala jalapensis* Casey are particular for this species, and the rest of its reproductive apparatus is very similar to that of other Melolonthidae [29]. Recently, Sanchez-Cruz et al. [30] reported that the internal walls of the genital chamber of *C. barrerai* females show oval or round 2–10 μm long bacteria. The male reproductive apparatus has been described for *C. jalapensis*, *Cyclocephala picta* Burmeister, *Cyclocephala sexpunctata* Castelnau, *Cyclocephala mafaffa* Burmeister, *Cyclocephala lunulata* Burmeister, and *Cyclocephala weidneri* Endrödi, with the parameres showing the widest variation among these species [9].

*Cyclocephala barrerai* is distributed in the central part of México, and it is considered a pest, as its larvae feed on economically important plants such as grasses and maize [31,32,33]. There is no available management program for this pest. However, adults are attracted to volatiles produced by symbiotic bacteria from the female genital chamber [30].

The study of the morphology of the reproductive structures will provide relevant information on the *C. barrerai* mating system, laying the foundations for applied research. This paper describes the gonadic maturation, the female and male genitalia, and some allometric relationships within each gender.

## 2. Materials and Methods

### 2.1. Cyclocephala barrerai Insects

*Cyclocephala barrerai* adults were collected at “La presa” baseball park at Puebla city, México (18°58′26.0″ N 98°14′46.4″ W), from May to June 2020. Collected insects were reared at the Chemical Ecology laboratory facilities of the Centro de Desarrollo de Productos Bióticos of the Instituto Politécnico Nacional and were kept at 26 ± 3 °C, 70 ± 20% HR and a 12L:12D light regime.

Insects were reared according to Sanchez-Cruz et al. (in preparation). In short, we setup 98 mating triads (a mature female and two mature males); each one was allocated to a 50 mL cylinder plastic container filled with 25 mL of soil and covered with a cheesecloth secured by a rubber band. Eggs were collected every 24 h and each one allocated to a 50 mL plastic container (the same one mentioned above) with a nutritious substrate (25 mL) until pupation. The substrate was kept humid by spraying tap water as needed. The nutritious substrate was replaced every month.

A single pupa was allocated to a 35 mL plastic container with 29 mL of humid soil. Four of these containers were kept in a 15 cm length by 5 cm height and 10 cm width squared container whose top had four holes for ventilation. To prevent pupae dehydration, 50 mL of water was added into the container. Newly emerged adults were collected 24 h after emergence and allocated to a 50 mL cylinder plastic container filled with 25 mL of soil in the above-mentioned plastic containers with the nutritious substrate.

### 2.2. Reproductive Apparatus Description

The reproductive apparatus (RA) of 20 virgin insects (10 for each gender) with no deformations or apparent injuries and bright cuticles of 18 d old were dissected in saline solution and stained for 5 min with 10% Giemsa. The RA was rinsed with tap water prior to observation on a Zeiss Discovery-V20 (Thornwood, NY, USA, EE. UU) stereoscopic microscope. Several papers [8,9,14] were used to identify the reproductive structures. Drawing and image processing were performed with Krita 4.4.2 software (Stichting Krita Foundation, Deventer, The Netherlands).

The ejaculatory bulb and genital capsule from males and the genital chamber, accessory glands, and ovaries from females were dehydrated by sequential immersion in ethanol at 70% (2 h), formaldehyde 40% (1 h), ethanol 90% (2 h), formaldehyde 40% (1 h), and 100% ethanol for (2 h) [34,35,36]. Microphotographs were obtained according to García-Hernández et al. [37], Delgado-Núñez et al. [38] for methodology. The structures were placed in aluminum stubs with double-sided carbon adhesive and viewed using an Environmental Electronic Scanning Microscope (ESEM) Model EVO LS10 Life Science (Carl Zeiss, Jena, Germany). The genital chamber and accessory glands were stained using 1% DAPI solution for 1 h. After remotion of DAPI by PBS solution, they were observed in a Laser Scanning Confocal Microscope (LSCM) model CLSM LSM 800 (Carl Zeiss, Jena, Germany).

### 2.3. Gonadic Maturation

To study the male gonadal maturation, 30 males were killed at 0, 10, and 18 d after emergence (*n* = 10), and their RA (accessory glands, vasa deferentia, and testes) were dissected and cleaned in saline solution. The RA were stained with 10% Giemsa for 5 min and observed under the stereoscope previously mentioned. The sperm were obtained from vasa efferentia and allocated to 1.5 mL Eppendorf tubes plus 200 µL of distilled water before manual stirring and observation was conducted under an ESEM.

To study the female gonadic maturation, the RA (ovaries and oocytes) of 10 females of 0, 6, 12, and 18 d old were dissected in saline solution, stained for 5 min with 10% Giemsa, and observed under the stereoscopic microscope; they were classified according to Grodowitz and Brewer [39]. The ovaries of 12 d old females were dissected to separate the ovarioles. The ovarioles were dehydrated at ethanol series (70%, 96%, and 100%) prior observation under the above-mentioned LSCM.

### 2.4. Allometry

All *C. barrerai* pupae were weighed using an analytical balance (Ohasus Explorer, 0.0001 g precision, Zurich, Switzerland). All newly emerged adults were weighted on the same scale and sexed according to the claw of the first legs where males present a larger claw than females. A total of 31 males and 36 females, 18 d old, were sacrificed by freezing.

To study the relationship between body parts and size, we separated the hind-tibia, middle-tibia, and the external-lamella of the antennae of both sexes and the genital chamber and *bursa copulatrix* in females and the tarsal claw (width and length) of the front legs of the male. All the structures were measured and photographed under a stereoscope Zeiss Discovery-V20 (Thornwood, NY, USA). The photos were analyzed with ImageJ software (Scion Corporation, Chicago, IL, USA) [40].

#### Statistical Analysis

The relationship between the body parts and the size or the length of the structures was analyzed by linear regression. We compared the measurements of the different body parts between sexes with a *t*-test. Unless stated otherwise, all reported data are the mean ± Standard Error of the Mean (SEM). All analyses were carried out on.

## 3. Results

### 3.1. Male Reproductive Apparatus

The male RA consists of two accessory glands; two glandular ducts; two testes, each with six follicles; six vasa efferentia; two vasa deferentia; two seminal vesicles; one ejaculatory duct; one ejaculatory bulb; one genital capsule; and one aedeagus (Figure 1). The accessory glands are long ducts with thick walls. The distal portion is long and narrow, and secretion is not observed in its lumen. The proximal portion is thicker and showed luminal secretion. The subsequent zone is thick with a fluid. The accessory glands are connected to the distal end of the ejaculatory duct (Figure 1). Each testis is composed of six free follicles, lobed, dorsoventrally flattened, in form of a “flower”, and every testis follicle is connected to vasa efferentia. The vasa efferentia are short and thin ducts connected to one vasa deferentia.

The vasa deferentia are long, with thicker walls, and the diameter of its anterior part is smaller than the posterior. Like the accessory glands, they open into the anterior end of the ejaculatory duct (Figure 1). One end of the ejaculatory duct is connected to the accessory glands and the vasa deferentia while the other connects to the ejaculatory bulb (Figure 1). The ejaculatory bulb is a membranous sac with fluted-like muscular tissue (Figure 2).

The endophallus is a membranous and large structure that connects to the ejaculatory duct (Figure 2A), and it is surrounded by the genital capsule (Figure 2B). The genital capsule is a symmetric structure formed of three parts (Figure 3):(1)The connecting membrane (Figure 3A) is in the ventral zone, connecting the phallobase to parameres.(2)The parameres (Figure 3A,B) are at the end of the genital capsule; they are symmetric, with a curvilinear triangle shape, and ornamented with small cones and pits (Figure 3C). Some areas at the proximal tip of the parameres have some pallid fine lines between the pits (Figure 3D).(3)Posterior phallobase and anterior phallobase (Figure 3E). The posterior phallobase is a sub-cylindric structure that involves the aedeagus and connects to the parameres. It is made up of one piece and presents a high diversity of ultrastructure (Figure 3F). The posterior phallobase has four types of ultrastructure: (1) large needle, (2) small needle, (3) pits (Figure 3F), and (4) small seta with a circular base (Figure 3G).

Newly emerged males of *C. barrerai* have an immature reproductive apparatus (Figure 4A). The accessory glands and vasa deferentia of newly emerged males are empty and white and clear with uniform walls. These structures from 10 d old males are enlarged and are partially filled with a white fluid.

When males are 18 d old, the distal section of the accessory glands are filled with a white content, and the mature sperm is in the proximal portion of the vasa deferentia. Mature sperm has a filamentary shape (Figure 4B).

### 3.2. Female Reproductive Apparatus

The female RA has a genital chamber, two pairs of genital plaques, two pairs of ventral plaques, a spermatheca with a spermathecal gland, a common oviduct, and two ovaries with six ovarioles each (Figure 5). The spermatheca is a tubular, sickle-shaped structure with a spermatheca muscle and spermatheca gland. The spermatheca gland is balloon-shaped and is larger than the spermatheca. Both structures join the spermathecal duct (Figure 5). The *bursa copulatrix* is a bimembranous and striated bag arising from the genital chamber. It is bigger than the common oviduct (Figure 5). The common oviduct is a large tubular structure connecting posteriorly to the anteroventral area of the genital chamber and bifurcating anteriorly to the paired lateral oviducts that connect to the paired ovaries (Figure 5).

The genital chamber is cylindric, in which the common oviduct, *bursa copulatrix*, and the spermatheca duct arrive (Figure 6A,B). The latest is made of muscular fibers rolled up over each other (Figure 6C,D). In a cross-section view (Figure 6D), the tissue is striated. The distal part, close to the genital plaques of the genital chamber, of 18 d old females has bacteria. These have a circular shape and are in the chamber inside the muscular tissue (Figure 6E,F). They can be found alone, aggregated, or in couples or triples. The bacteria size varied from 3 to 14 μm. The genital chamber has two types of genital plates:(1)Dorsal plates are sclerotized rectangular shape structures. They insert near the accessory glands, around the genital chamber.(2)Ventral plates are oval with many setae and inserted in the distal part of the genital chamber (Figure 7A). The setae interact with the parameres of the male during mating. The ventral part has pits (Figure 7B).

We observed two types of accessory glands:(1)Type I is oval-shaped, and it inserts in the genital chamber (Figure 5).(2)Type II is sclerotized (Figure 8A) and has a spherical shape. It is formed by acellular triangular plates (Figure 8B–D).

Females of *C. barrerai* have two telotrophic ovaries, each with six ovarioles (Figure 9). Ovarioles are composed of germarium and vitellarium, while mature oocytes are in the oviduct (Figure 9A,B). The germarium contains the germinative cells that will produce gametes (oogonia) (Figure 9C,D). The vitellarium contains the immature oocytes (ovarian follicles) (Figure 9E).

The ovarian development of *C. barrerai* (Figure 10) exhibits three nulliparous stages and a parous (with ovulation) one (Figure 10). On day 0 (emerged from pupae), the ovary is in nullipara 1 stage; germarium and vitellarium are not differentiated and are without follicles. On day six, the ovary is in nullipara 2 stage: well-defined ovarioles with multiple follicles, the proximal follicle transparent indicating that it had not completed maturation. On day 12, the ovary is in nullipara 3 stage: proximal follicle opaquer and closer to ovulation. On day 12, the ovary is in the parous stage.

### 3.3. Allometry in Cyclocephala barrerai

#### 3.3.1. Relationships between Body Parts and Weight

Measures of the different body parts used in this study are presented in Table 1. Lab-reared *Cyclocephala barrerai* has a linear and positive relationship among pupal weight and female weight (R^2^ = 0.820, F = 254.2, df = 1, 57, *p* < 0.001) and male weight (R^2^ = 0.833, F = 279.531, df = 1, 57, *p* < 0.001). Female weight and her middle-tibial length (R^2^ = 0.566, F = 44.28, df = 1, 35, *p* < 0.001, Figure 11A) and hind-tibial length (R^2^ = 0.437, F = 25.584, df = 1, 34, *p* < 0.001, Figure 11B) have linear and positive relationships. Similar relationships occur among male weight and his middle-tibial length (R^2^ = 0.566, F = 44.28, df = 1, 35, *p* < 0.001, Figure 11C) and his hind-tibial length (R^2^ = 0.437, F = 25.584, df = 1, 34, *p* < 0.001, Figure 11D) were observed.

The forelegs have two claws which were named as tarsal claw 1 and tarsal claw 2 (Figure 12A). The tarsal claws of the forelegs in males are thicker than the other claws (Figure 12B). The underside of the tarsal claw 1 has an opening with channels, and there is a protuberance at the tip of the claw (Figure 12C). Tarsal claw 2 is thinner than the first one and has no evident structures except from some light lines along the claw (Figure 12D).

Lab-reared *Cyclocephala barrerai* male weight and lamella length (R^2^ = 0.772, F = 60.842, df = 1, 19, *p* < 0.001, Figure 13A) and lamella area (R^2^ = 0.512, F = 18.867, df = 1, 19, *p* < 0.001, Figure 13B) have a linear and positive relationship. Similar relationships occur among male weight and the width of the base of their tarsal claw 1 (R^2^ = 0.605, F = 21.472, df = 1, 16, *p* < 0.001, Figure 13C) and tarsal 1 claw length (R^2^ = 0.432, F = 10.662, df = 1, 16, *p* < 0.001, Figure 13D).

Lab-reared *Cyclocephala barrerai* female body weight and genital chamber area (R^2^ = 0.696, F = 32.05, df = 1, 15, *p* < 0.001, Figure 14A), *bursa copulatrix* length (R^2^ = 0.185, F = 5.678, df = 1, 26, *p* = 0.025, Figure 14B), lamella length (R^2^ = 0.615, F = 30.334, df = 1, 19, *p* < 0.001, Figure 14C), and lamella area (R^2^ = 0.474, F = 12.619, df = 1, 15, *p* = 0.003, Figure 14D) have a linear and positive relationship.

There was no relationship between lab-reared *C. barrerai* male weight and the width of the base of tarsal claw 2 (R^2^ = 0.101, F = 1.456, df = 1, 15, *p* = 0.249) and tarsal claw 2 length (R^2^ = 0.167, F = 2.610, df = 1, 15, *p* = 0.130). No relationship was observed between lab-reared female weight and the *bursa copulatrix* area (R^2^ = 0.0702, F = 1.813, df = 1, 25, *p* = 0.191) and the genital chamber length (R^2^ = 0.0546, F = 0.983, df = 1, 18, *p* = 0.335).

#### 3.3.2. Sexual Dimorphism in Lab-Reared Adults

The weight of pupae that produced females did not statistically differ from pupae that developed into males (t = 1.839, df = 57, *p* = 0.069). However, at emergence, adult females were heavier than adult males (t = 2.112, df = 57, *p* = 0.037, Figure 15A). The lamella length (t = 20.796, df = 39, *p* < 0.001, Figure 15B) and the lamella area (t = 18.632, df = 34, *p* < 0.001, Figure 15C) of lab-reared adult males were bigger than that of females.

## 4. Discussion

The male reproductive apparatus of *C. barrerai* has a similar arrangement and organization of the Melolonthidae: testes, accessory glands, and vasa efferentia and vasa deferentia [9,11,13]. Like the six *Cyclocephala* species described by Martínez et al. [9], the male reproductive apparatus lacks a glandular reservoir, and the ejaculator duct and ejaculatory bulb present well-developed muscular fibers. The spermatophore is formed at the two latter structures by fusing the sperm and the content of the accessory glands before it is transferred to the females [2].

*C. barrerai* male gonads mature after emergence despite those adults not feeding and are sexually active. This situation has also been reported for *Paraglenea fortunei* Saunders (Cerambycidae) [41], *Hylobius pales* Herbst (Curculionidae) [42], and *Oemona hirta* Fabricius (Cerambycidae) [43].

Due to their taxonomic relevance, the parameres of *Cyclocephala* have been extensively described. Breeschoten et al. [44] analyzed and compared the parameres of 48 species identifying three different types: (1) symmetric, (2) minor asymmetry, and (3) asymmetric. The same authors indicated that species subject to higher interspecific selection pressure, such as sympatric species of the same genera, have asymmetric parameres. Males of *C. barrerai* have a genital capsule type 1 formed for three plates with symmetric parameres, indicating that this attribute is not under interspecific selection pressure, as *C. barrerai* is the only one inhabiting the collecting area.

The ultrastructure found at the phallobase and parameres are like the trichoid and basiconic sensilla, respectively; both sensilla have been reported on the antennae of the Melolonthidae [45,46]. Ultrastructures similar to those found in *C. barrerai* have been reported in the genital capsule of species of the Chrysomelidae, but no explanation of their diversity or function has been proposed so far [47,48,49].

We propose that the ultrastructure at the genital capsule of *C. barrerai* has a mechanoreceptor function facilitating sexual recognition and coupling, as proposed for *Drosophila melanogaster* Meigen [50]. This hypothesis also explains the function of the setae of the ventral genital plaques found on *C. barrerai* females as the genital plaques and the genital capsule have an active role during mating [12]. The parameres of the male provide support during mating and in conjunction with the female genital plaques, facilitate the insertion of the penis and sperm transfer [51].

The reproductive apparatus of *C. barrerai* females has the general arrangement of the Melolonthidae [8,13,14,29]. The *bursa copulatrix* and the spermatheca connect to the genital chamber, storing and nourishing the sperm inside the female body until oviposition [52]. The genital chamber presents muscular fibers as mentioned for *C. lunulata* (Benítez-Herrera, unpublished data), and searching for pheromone glandular cells is mandatory.

The female genital chamber has muscular fibers, and its static positive allometry indicates a female cryptic sexual selection process. This provides the female some control over sperm transfer to avoid mating to unfit individuals and to reduce the risk of hybridization [21,53]. The parameres’ morphology is the result of the coevolution of the male and female genitalia [12,21,42,54].

The presence of microorganisms in the female genital chamber agrees with a previous report [30] that identified *Klebsiella oxytoca* and *Citrobacter freundii* in *C. barrerai* wild females. The presence of microorganisms in lab-reared females indicates that these might be transferred by the mother or acquired during the larval period and not transferred by the male during mating. The architecture of the genital chamber and the presence of microorganisms suggest that they are relevant during attractant production [14,30].

*Cyclocephala barrerai* and *C. jalapensis* have type II accessory glands. These are highly sclerotized superimposed plaques [29] whose function is unknown. This contrasts with those observed in *Phyllophaga* [14], *Macrodactylus* [8], and *C. zealandica* [16], where the type II accessory glands are semi-sclerotized, and the cells form conduits to transport pheromones.

*Cyclocephala barrerai* female and male gonadic maturation initiate once the adults emerge from the pupae, and they surface and search for mates when they are sexually mature: similar behavior to that reported for *Holotrichia titanis* Reitter, *Holotrichia trichopora* Fairmaire, and *Anomala exoleta* Faldermann [15]. *C. barrerai* adults from the lab colony were not fed, and their digestive system was filled with a dark liquid. Most Coleoptera adults depend (partially or completely) on food for gonadic maturation [55,56,57,58]. *C. barrerai* must obtain and store all the nutrients needed for metamorphosis and gonadic maturation as larvae, explaining their voracious appetite as larvae [58].

There is a strong relationship between the hind-tibia length and the body weight in *C. barrerai* and can be used to assess their bodyweight [59]. *C. barrerai* females are heavier than males; however, in *C. borealis*, it is the other way around: males are heavier than females [25]. This could be related to the life history of each species. In addition, males’ lamellae have a larger surface and are longer than those of females. According to the drawings of the males’ lamellae for *Cyclocephala borealis* Arrow by Johnson [28] and for *Cyclocephala signaticollis* Burmeister by Carne [60], males’ are larger than those of females. Unfortunately, there is no quantitative information on lamella length and surface area reported for any other *Cyclocephala*.

When mate location relies on the prompt detection of pheromones, those individuals with larger antennae or more sensitive sensilla have a clear advantage over the rest of the sexually active population. Males of *C. barrerai* present static positive allometry due to their strong relationship between antenna length and body size [61] indicating a strong sexual selection process [21,62]; therefore, heavier males have an adaptative advantage over lighter males during mate location and mating [63,64]. Mate location on *Cyclocephala* depends on the prompt detection of receptive females in the vicinity, as a contest between males is rare.

It is possible to observe delicate ultrastructure under the advance Life Science microscopes. Samples do not require extensive handing or manipulation, so results are relievable, replicable, and comparable among studies [65,66]. The ultrastructure observed in the RA helps to explain their possible role during pairing.

## 5. Conclusions

The reproductive apparatuses of *C. barrerai* show the same arrangement and organization as the Melolonthidae, with ultrastructures in the phallobase and the parameres. The genital chamber has sclerotized ventral genital plates and type II accessory glands. Male and female gonadic maturation is achieved 18 d after emergence, and no food intake is required. The hind-tibia length and lamella length can be used to assess female and male bodyweight. The positive relationship between the size of the first-leg claw and antennae and the weight suggests that those structures had been formed under sexual selection. The use of new microscopy techniques offered evidence of the ultrastructure of the reproductive apparatus relevant during mating, providing useful information for the development of behavioral control techniques.

## Figures and Tables

**Figure 1 insects-13-00638-f001:**
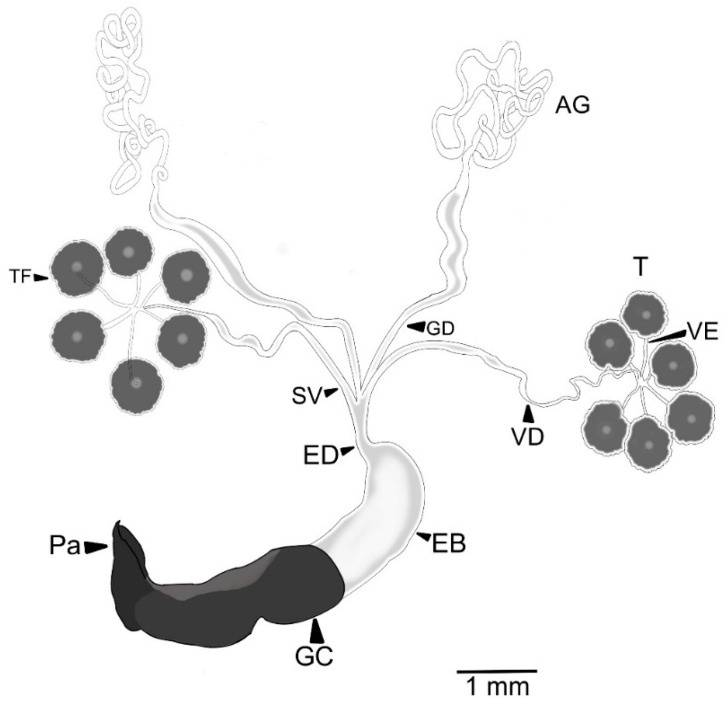
Reproductive apparatus of 18 d old *Cyclocephala barrerai* male. AG = accessory glands, GD = glandular duct, T = testes, TF = testis follicles, Vd = vasa deferentia, Ve = vasa efferentia, Sv = seminal vesicle, ED = ejaculatory duct, EB = ejaculatory bulb, GC = genital capsule, Pa = parameres.

**Figure 2 insects-13-00638-f002:**
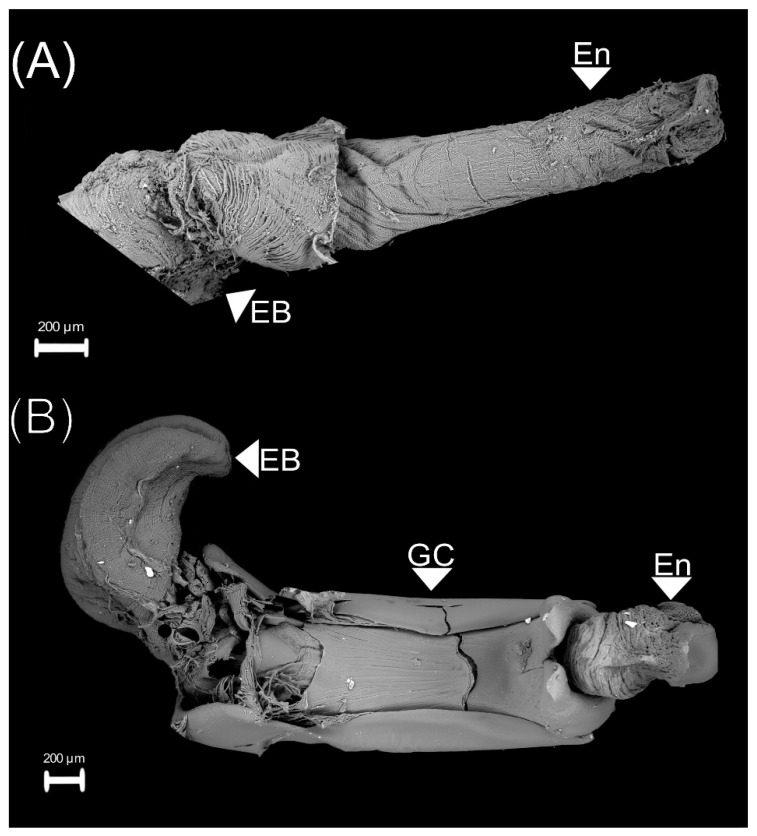
Endophallus of 18 d old *Cyclocephala barrerai* male. (**A**) endophallus and ejaculatory bulb, (**B**) genital capsule and endophallus. En = endophallus, EB = ejaculatory bulb, GC = genital capsule.

**Figure 3 insects-13-00638-f003:**
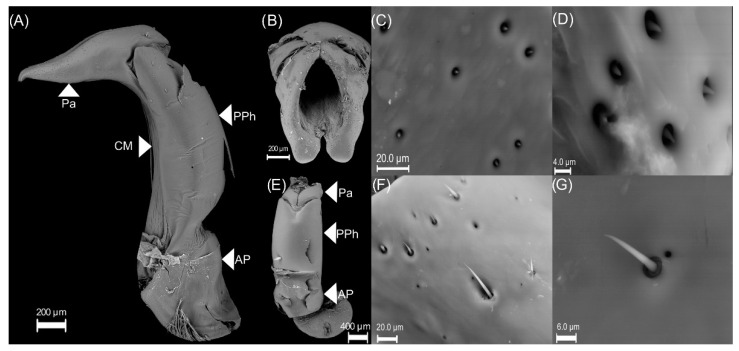
Genital capsule of 18 d old *Cyclocephala barrerai* male. (**A**) genital capsule (lateral view), (**B**) parameres (frontal view), (**C**) ultrastructure in parameres (distal view), (**D**) ultrastructure in parameres (proximal view), (**E**) genital capsule (dorsal view), (**F**) pits and ultrastructure in phallobase (proximal view), (**G**) pits and ultrastructure in phallobase (posterior view). Pa = parameres, PPh = posterior phallobase, CM = connecting membrane, AP = anterior phallobase.

**Figure 4 insects-13-00638-f004:**
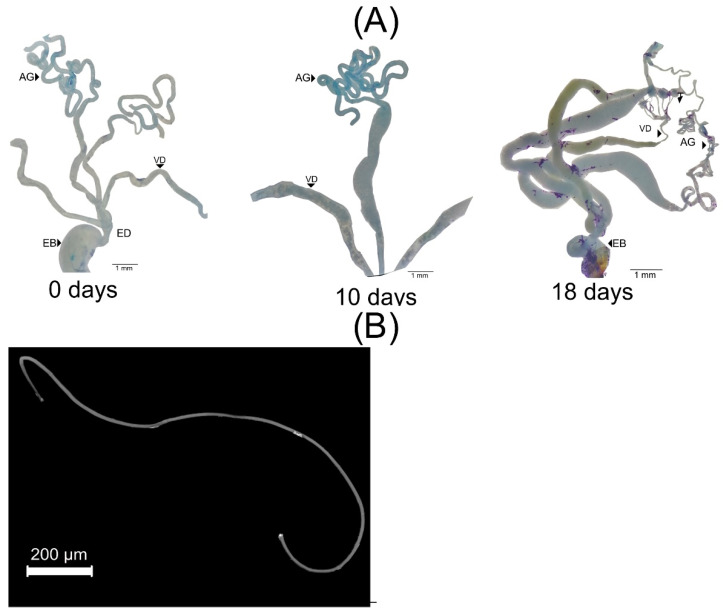
Development of male reproductive apparatus and sperm of *Cyclocephala barrerai*: (**A**) development through time, (**B**) sperm from an 18 d old male. AG = accessory glands, Vd = vasa defferentia, ED = ejaculatory duct, EB = ejaculatory bulb.

**Figure 5 insects-13-00638-f005:**
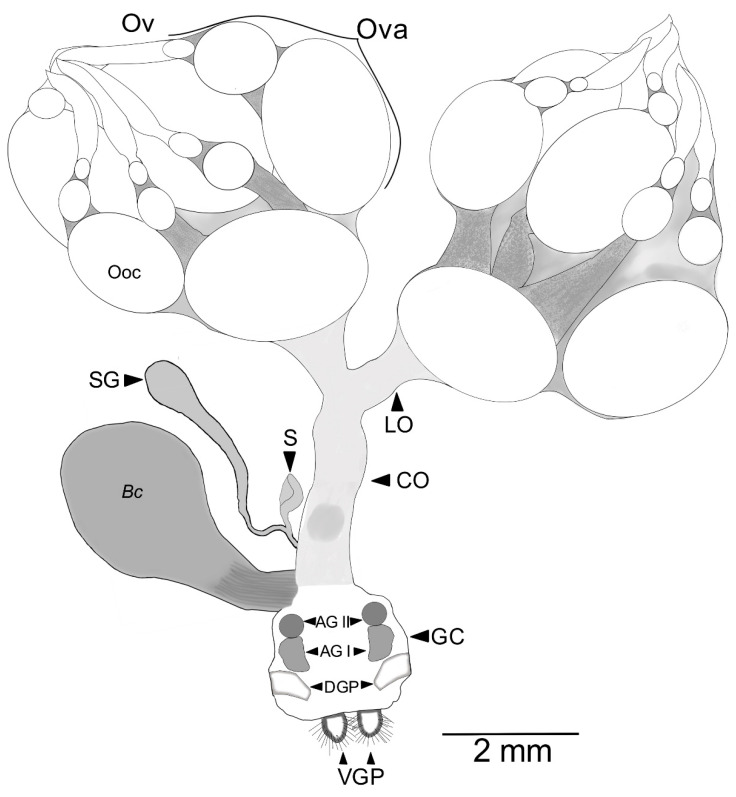
Reproductive apparatus of 18 d old virgin *Cyclocephala barrerai* female. Bc = *bursa copulatrix*, GC = genital chamber, S = spermatheca, SG = spermathecal gland, AG I = accessory gland type I, AG II = accessory gland type II, Grm = gremarium, Ooc = oocyte, Ov = ovary, Ova = ovariole, CO = common oviduct, LO = lateral oviduct, DGP = dorsal genital plates, VGP = ventral genital plates.

**Figure 6 insects-13-00638-f006:**
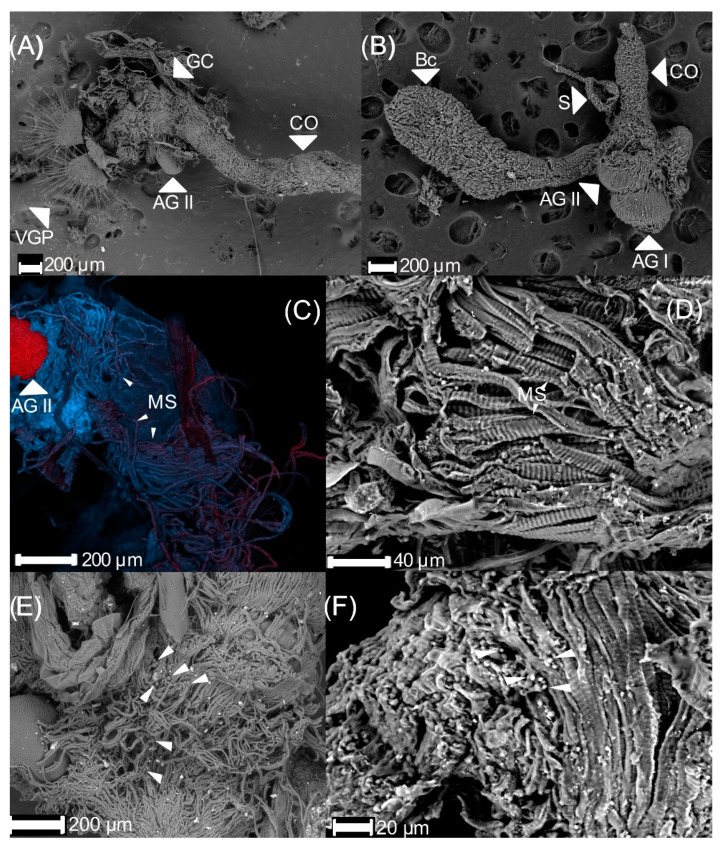
Genital chamber of 18 d old virgin *Cyclocephala barrerai* female: (**A**) genital chamber (interior view), (**B**) genital chamber (exterior view), (**C**) muscular structure of genital chamber (confocal scanning laser microscopy view), (**D**) cross-section, (**E**) bacteria in the genital chamber (white arrows), (**F**) bacteria into the muscular structure of the genital chamber (white arrows). GC = genital chamber, CO = common oviduct, AG = accessory glands type II, VGP = ventral genital plates, MS = muscular structures.

**Figure 7 insects-13-00638-f007:**
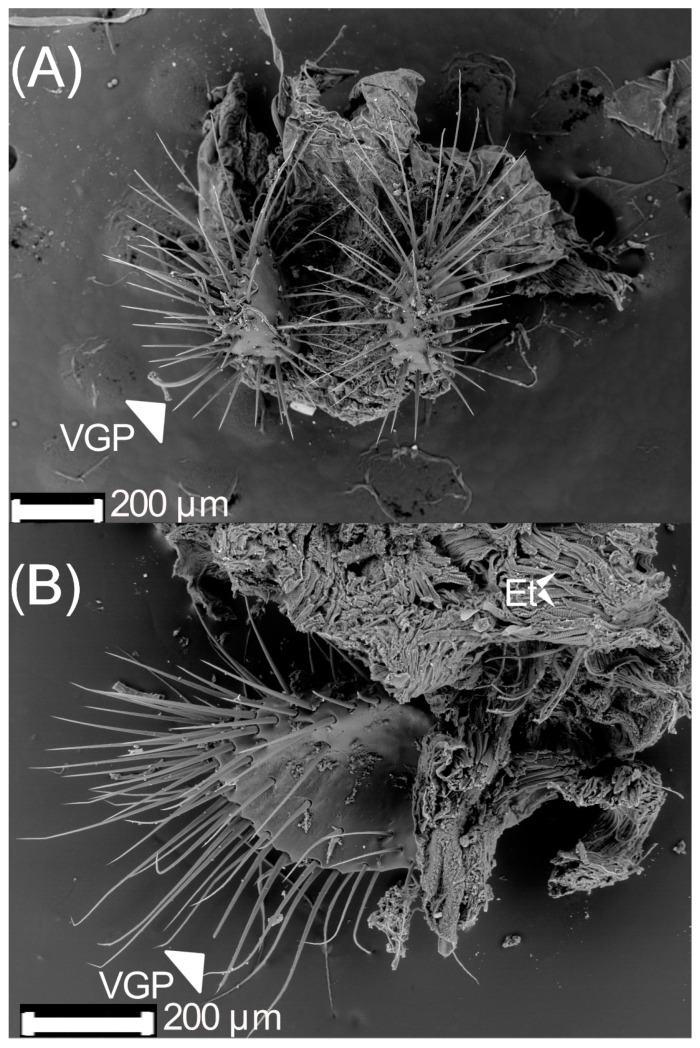
Ventral genital plates of 18 d old *Cyclocephala barrerai* virgin female: (**A**) ventral genital plates (front view), (**B**) ventral genital plates (lateral view). VGP = ventral genital plates.

**Figure 8 insects-13-00638-f008:**
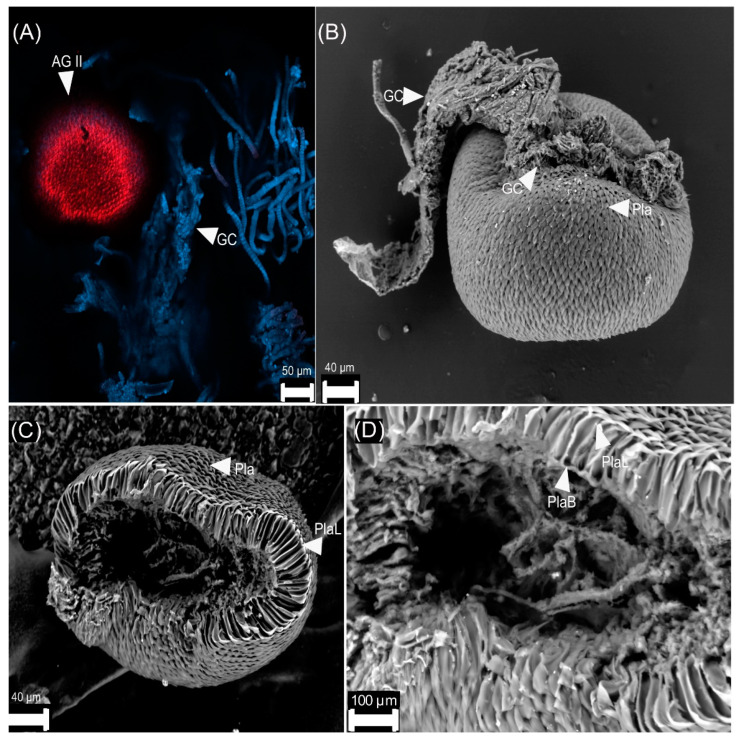
Accessory glands type II of *Cyclocephala barrerai* female: (**A**) tissue in blue and keratinized structure in red (confocal scanning laser microscopy), (**B**) accessory gland type II with tissue of the genital chamber, (**C**) accessory gland type II (internal view), (**D**) plates in the accessory gland type II (close-up), AG II = accessory glands type II, GC = genital chamber, Pla = plates, PlaL = plates (lateral view), PlaB = plates (ventral view).

**Figure 9 insects-13-00638-f009:**
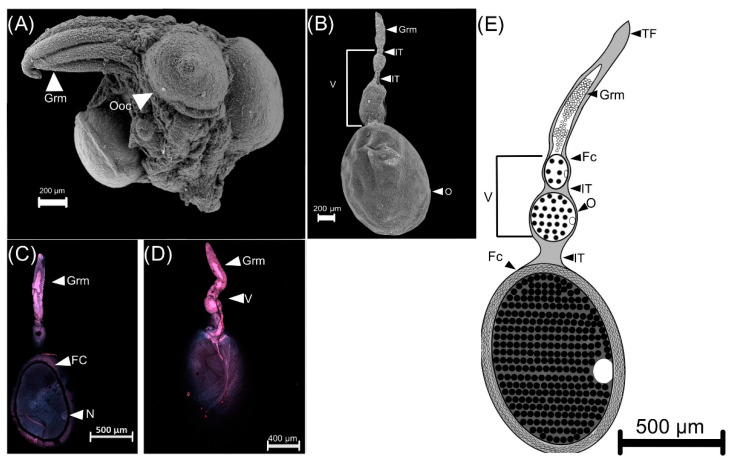
Telotrophic ovary of *Cyclocephala barrerai*. (**A**) ovary of 6 d old female, (**B**) ovariole of 12 d old female, (**C**,**D**) ovarioles (confocal scanning laser microscopy view). (**E**) scheme of ovary. Grm = gremarium, Ooc = oocyte, Fc = Follicular cells, V = vitellarium, TF = terminal filament, N = nucleus, IT = interfollicular tissue.

**Figure 10 insects-13-00638-f010:**
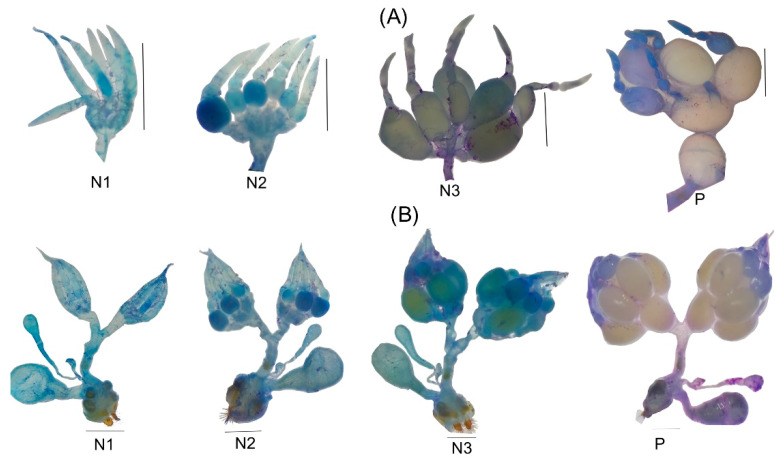
Ovarian development of *Cyclocephala barrerai* female. (**A**) individual ovariole maturation, (**B**) ovary maturation N1 = nulliparous 1, N2 = nulliparous 2, N3 = nulliparous 3, P = parous. Bars = 2 mm.

**Figure 11 insects-13-00638-f011:**
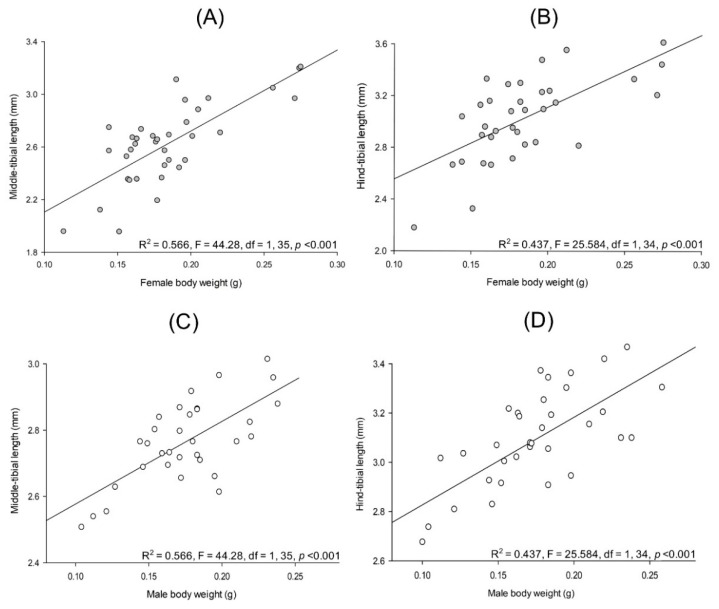
Relationships between body weight and tibial length on *Cyclocephala barrerai*. Female body weight and (**A**) middle-tibial length and (**B**) hind-tibial length; male body weight and (**C**) middle-tibial length and (**D**) hind-tibial length. Female = filled dots, male = open dots.

**Figure 12 insects-13-00638-f012:**
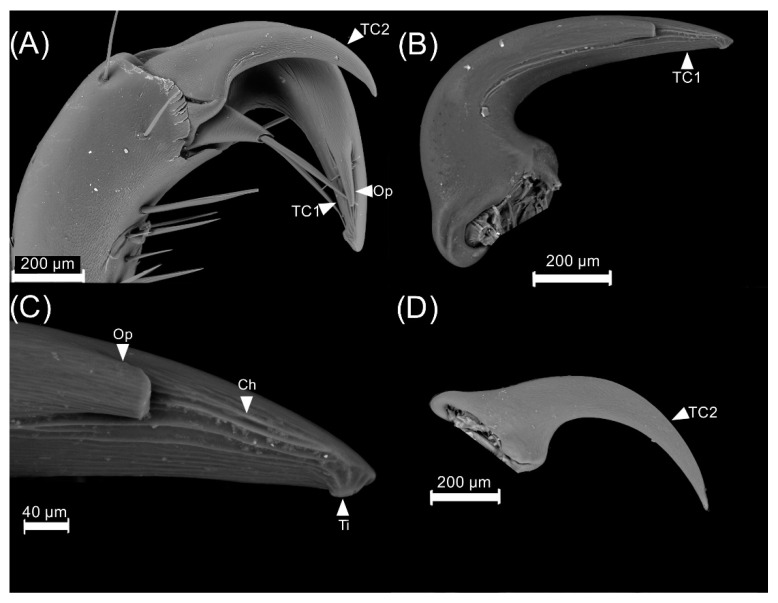
Tarsal claw of the first pair of legs of *Cyclocephala barrerai* males: (**A**) general view of the tarsal claw composition, (**B**) tarsal claw 1, (**C**) close-up of tarsal claw 1, (**D**) tarsal claw 2. TC1 = tarsal claw 1, TC2 = tarsal claw 2, Op = opening, Ch = channel, Ti = tip.

**Figure 13 insects-13-00638-f013:**
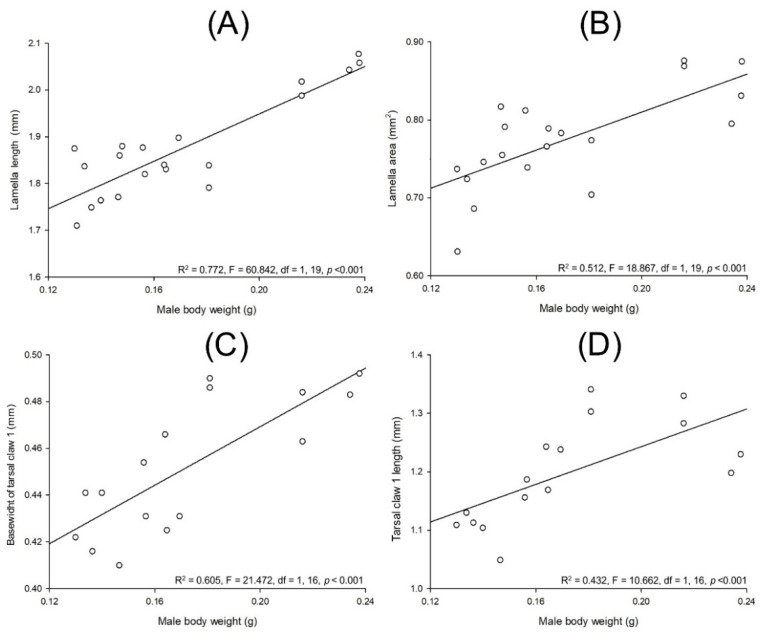
Relationships between body weight and different body parts on *Cyclocephala barrerai* males: body weight and (**A**) lamella length, (**B**) lamella area, (**C**) width of the base of tarsal claw 1, and (**D**) tarsal claw 1 length.

**Figure 14 insects-13-00638-f014:**
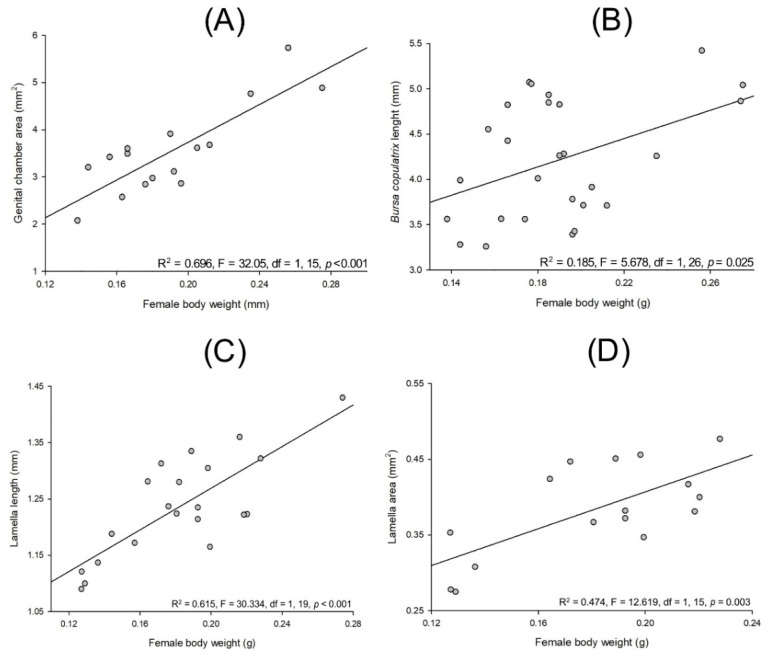
Relationship between body weight and body parts on *Cyclocephala barrerai* females: body weight and (**A**) genital chamber area, (**B**) *bursa copulatrix* length, (**C**) lamella length and (**D**) lamella area.

**Figure 15 insects-13-00638-f015:**
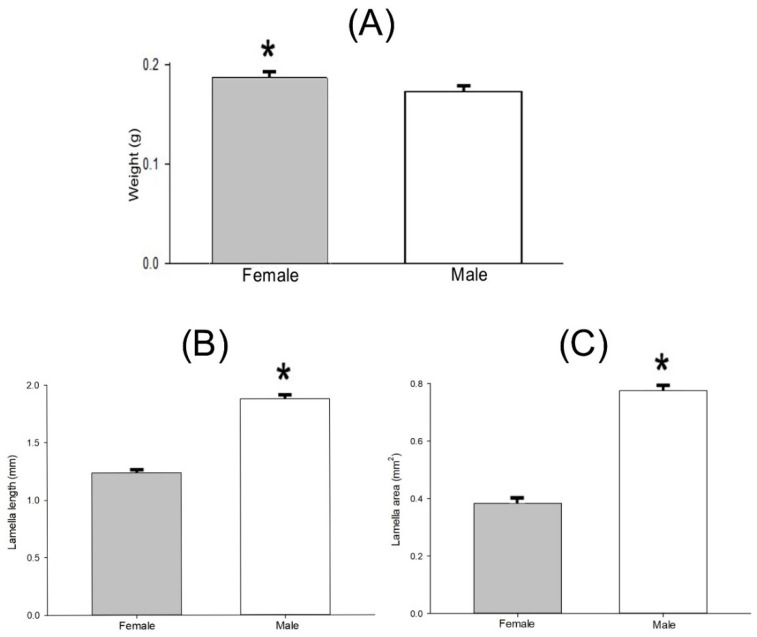
Mean ± SEM of weight (**A**), lamella length (**B**), and lamella area (**C**) of lab-reared males and females of *Cyclocephala barrerai* adults. *t*-test, * = significance 0.05.

**Table 1 insects-13-00638-t001:** Measurement of different body parts of *Cyclocephala barrerai* adults.

Structure	Lab-Reared Adults Mean ± SEM (*n*)	
	Female	Male
Pupa weight	0.308 ± 0.006 g (58)	0.291 ± 0.006 g (58)
Adult weight	0.188 ± 0.004 g (58)	0.175 ± 0.004 g (58)
Middle-tibial length	2.625 ± 0.051 mm (36)	2.770 ± 0.022 mm (33)
Hind-tibial length	3.025 ± 0.054 mm (35)	3.077 ± 0.048 mm (36)
Lamella length	1.236 ± 0.089 mm (21)	1.876 ± 0.024 mm (20)
Lamella area	0.383 ± 0.015 mm^2^ (16)	0.775 ± 0.014 mm^2^ (20)
Tarsal claw 1 length	-	1.199 ± 0.0218 mm (16)
Tarsal claw 1 width base	-	0.452 ± 0.007 mm (16)
Tarsal claw 2 length	-	0.839 ± 0.065 mm (15)
Tarsal claw 2 width base	-	0.0388 ± 0.008 mm (15)
Genital chamber length	1.993 ± 0.109 mm (19)	-
Genital chamber area	3.875 ± 0.291 mm^2^ (16)	-
*Bursa copulatrix* length	4.223 ± 0.145 mm (27)	-
*Bursa copulatrix* area	5.089 ± 0.357 mm^2^ (26)	-

Standard error of mean = SEM.

## Data Availability

All the data presented in this study are in the manuscript.

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
