# Peer review of "Reproductive Apparatus, Gonadic Maturation, and Allometry of Cyclocephala barrerai Martínez (Coleoptera: Melolonthidae: Dynastinae)"

_insects, 2022, doi:10.3390/insects13070638_

Round 1

Reviewer 1 Report

The manuscript Reproductive apparatus, gonadic maturation, and allometry of Cyclocephala barrerai Burmeister (Coleoptera: Melolonthidae: Dynastinae)”, by Sanchez-Cruz et al., deals with a subject that is important for both reproductive biology and systematics. From my point of view, it can only be accepted for publication after it undergoes significant changes. For example, I strongly suggest that the result be written as a running text, not as a large number of small and unnecessary paragraphs. Nevertheless, many paragraphs are just quotes from previously published works, which is very unusual. It is also necessary that the manuscript undergo a careful and extensive language review (English). Finally, I suggest the authors look carefully at the many suggestions and comments I have made on the pdf.

Reviewer 2 Report

The paper provides interesting data on the reproductive structures of a species of economic importance. In general, I recommend the study to be published in the MDPI Insects after minor revision. First of all, I recommend to verify English of the paper, as several language errors are present in the current version of the text. The Authors consequently use the noun "nail" instead of "claw" - this shall be corrected within the text.  In the Material & methods section, it would be good to include more information on: the method used to confirm that structures visible on the Fig. 6E, F are bacteria indeed. Moreover the authors refer to "healthy" insects - how were the  individuals examined to verify it they were "healthy" and what is the precise meaning of this term? In some parts abbreviations are either incorrect or not explained, all these cases are marked on the manuscript file which is enclosed. I am also not convinced if there is a need to refer in the section "Results" to definitions of particular structures (see the sections 3.2.2. and 3.2.3). For minor comments and corrections, see the enclosed pdf file.

Reviewer 3 Report

The work is interesting because it deals with a morphological description of the reproductive system and stage of sexual maturation of an insect species considered a pest in Mexico. However, there are some points that must be clarified:

Summary:

It is not possible to identify the objectives of the work. Furthermore, a conclusion is lacking.

Introduction:

Authors should further explore the species studied and emphasize how studies with microscopy and allometry can contribute to applied research.

Material and methods:

2.2. Reproductive apparatus description: Lines 132-135. What is the purpose of using confocal only for the genital chamber and accessory glands?

2.3. Male gonadic maturation: Line 142: “200 μm of dissolved water”????

Line 146: What is the criterion for choosing females with 12 days to remove their ovarioles?

2.4. Allometry: Please provide the number of pupae, emerged adults and analyzed photos.

Discussion:

Need a conclusion

Round 2

Reviewer 1 Report

Reading the manuscript carefully, I noticed that the authors consider virtually every suggestion that has been made. So I suggest it be accepted for publication after the three suggestions/comments below on methodology have been considered.

Lines 123-126:

Comment: I would like the authors to justify using aqueous paraformaldehyde during dehydration, even between 90% and 100% alcohol; this makes no sense. Furthermore, this procedure was performed in none of the two studies [34, 35] cited by the authors. PS. The concentration of formaldehyde (formalin) to fix biological materials is 2.0-2.5%, not 40%.

Lines 130-133:

From: “The genital chamber and accessory glands were stained using 1% DAPI solution for 1 h prior to remotion of DAPI by 10% PBS solution. The genital chamber and accessory glands were observed in a Laser Scanning Confocal Microscope (LSCM) model CLSM LSM 800 (Carl Zeiss, Germany).”

Change to: The genital chamber and accessory glands were stained using 1% DAPI solution for 1h. After remotion of DAPI by PBS solution, they were observed in a Laser Scanning Confocal Microscope (LSCM) model CLSM LSM 800 (Carl Zeiss, Germany).

Lines 142-143:

Why above (line 118) you used Giemsa at 1% and here at 10%?
